# Classification of Benign–Malignant Thyroid Nodules Based on Hyperspectral Technology

**DOI:** 10.3390/s24103197

**Published:** 2024-05-17

**Authors:** Junjie Wang, Jian Du, Chenglong Tao, Meijie Qi, Jiayue Yan, Bingliang Hu, Zhoufeng Zhang

**Affiliations:** 1Xi’an Institute of Optics and Precision Mechanics, Chinese Academy of Sciences, Xi’an 710119, China; wangjunjie@opt.ac.cn (J.W.); dujian@opt.ac.cn (J.D.); taochenglong@opt.ac.cn (C.T.); mjqi@stu.xidian.edu.cn (M.Q.); yanjiayue21@mails.ucas.ac.cn (J.Y.); 2University of Chinese Academy of Sciences, Beijing 100049, China; 3Key Laboratory of Biomedical Spectroscopy of Xi’an, Xi’an 710119, China

**Keywords:** hyperspectral image, thyroid nodules, classification, spectral characteristics

## Abstract

In recent years, the incidence of thyroid cancer has rapidly increased. To address the issue of the inefficient diagnosis of thyroid cancer during surgery, we propose a rapid method for the diagnosis of benign and malignant thyroid nodules based on hyperspectral technology. Firstly, using our self-developed thyroid nodule hyperspectral acquisition system, data for a large number of diverse thyroid nodule samples were obtained, providing a foundation for subsequent diagnosis. Secondly, to better meet clinical practical needs, we address the current situation of medical hyperspectral image classification research being mainly focused on pixel-based region segmentation, by proposing a method for nodule classification as benign or malignant based on thyroid nodule hyperspectral data blocks. Using 3D CNN and VGG16 networks as a basis, we designed a neural network algorithm (V3Dnet) for classification based on three-dimensional hyperspectral data blocks. In the case of a dataset with a block size of 50 × 50 × 196, the classification accuracy for benign and malignant samples reaches 84.63%. We also investigated the impact of data block size on the classification performance and constructed a classification model that includes thyroid nodule sample acquisition, hyperspectral data preprocessing, and an algorithm for thyroid nodule classification as benign and malignant based on hyperspectral data blocks. The proposed model for thyroid nodule classification is expected to be applied in thyroid surgery, thereby improving surgical accuracy and providing strong support for scientific research in related fields.

## 1. Introduction

In recent years, cancer has gradually become a focus of public attention. Among the various types, thyroid cancer has received widespread attention as the most common endocrine malignancy [1]. The incidence of thyroid cancer has been rising since the 1980s, particularly in the past 20 years, where the incidence rate has tripled. Unfortunately, in the past 10 years, the incidence rate of thyroid cancer in China has increased more than five times. Although the 5-year survival rate for thyroid cancer is relatively high, it is significantly decreased if not treated promptly, with fewer than 60% of patients surviving, and furthermore, the risk of lymph node metastasis is over 50%. Therefore, the prevention, diagnosis, and treatment of early thyroid cancer are crucial to its detection as well as preventing patient deterioration.

With the continuous progress of medical imaging technology, image-based medical analysis has gradually become a research hotspot. Meanwhile, thanks to the rapid development in fields such as image processing, artificial intelligence, and pattern recognition, the use of computer image processing technology to analyze and process images of lesion areas is also receiving increasing attention [2,3,4]. The advancement of medical image processing not only results in the significantly reduced subjectivity of medical diagnosis, thereby effectively reducing the incidence of missed and misdiagnosed cases, but also significantly enhances the working efficiency of doctors. At the same time, this also saves patients much time and economic costs, reducing their burden from the diagnostic process.

At present, ultrasound imaging technology and diagnostic techniques based on pathological images are particularly eye-catching. Ultrasound imaging has been applied in the diagnosis of thyroid nodules based on its advantages of being fast and non-invasive, However, due to its low imaging resolution, only a preliminary diagnosis of thyroid nodules is possible, and it is not possible to determine their malignancy, both of which represent limitations for application in thyroid diagnosis. Pathological diagnosis is considered the gold standard for the accurate diagnosis, treatment, and prevention of thyroid cancer. This method demonstrates the best performance in terms of both specificity and sensitivity. Histopathological images usually have high resolution and fine details, thereby accurately depicting the geometric structure of organs and the complex texture features of cellular lesions [5]. This method for in-depth and meticulous observation enables pathologists to comprehensively understand the overall changes in organs and the characteristics of cellular lesions, resulting in more accurate diagnostic results. Compared with methods based on ultrasound images, pathological diagnosis is more objective and accurate. Artificial intelligence methods were used in the work of Xi et al. to improve the diagnosis of thyroid cancer, demonstrating a certain degree of ability in prediction [6].

Although pathological diagnosis can be used to effectively distinguish between benign and malignant thyroid nodules in the diagnosis of thyroid cancer, the process is quite complex. To obtain an accurate diagnosis, pathologists need to perform a series of treatments on tissue samples, in addition to conducting detailed histopathological examinations under a high-powered microscope. Such work is dependent on pathologists possessing an extremely high level of professional knowledge and clinical experience, which is difficult to achieve in the primary healthcare system. In addition, doctors may be affected by fatigue and subjective judgment during pathological diagnosis, which increases the risk of misjudgment and omission. Especially during surgery, the timely diagnosis of thyroid lesions is crucial, and a long time is often required to obtain more accurate results from pathological sections, which undoubtedly increases the risk and uncertainty of surgery.

In response to the shortcomings of pathological sectioning in intraoperative applications, tissue classification technology based on hyperspectral images has demonstrated significant advantages. Hyperspectral imaging technology was first applied in the field of remote sensing, achieving the synchronous acquisition of multi-dimensional information of surface features by obtaining much continuous spectral information [7]. It has been widely used in the fields of geological remote sensing, agricultural vegetation, and military reconnaissance among others [8]. Similarly, various spectral imaging systems have also been developed for evaluating various biological organs and tissues [9]. When lesions or carcinogenesis is present in biological tissues, it can be observed as changes in the cellular morphology and spatial texture in the pathological images of these tissues [10]. In addition, there will be corresponding changes in the tissue organization and chemical composition. These changes are reflected in the spectral information of substances in terms of their composition and content, such as through the shape of spectral lines and absorption peaks which provides the possibility for applying hyperspectral imaging technology in the classification of thyroid nodules during surgery [11]. The classification of hyperspectral images using neural network models is very rapid and can thus fully meet the clinical needs related to the intraoperative diagnosis of thyroid nodules.

It has been proven, in practice, that hyperspectral technology has unique advantages in rapid medical diagnosis. At present, medical hyperspectral image classification is becoming increasingly popular in medical diagnostic applications. High-resolution hyperspectral images can provide richer spectral features, thereby increasing the accuracy of classification tasks. This technology has mainly been used for the classification of cancer tissues and cells. Traditionally, machine learning methods have often been used for the classification of medical hyperspectral images. For example, Torti et al. first used a supervised classification algorithm consisting of principal component analysis (PCA), support vector machine (SVM), and K-nearest neighbor (KNN) for classification [12]. Fabelo et al. used a combination of supervised and unsupervised methods, using SVM for supervised pixel classification, and then dimensionality reduction algorithms to process the data [13]. Wang et al. proposed a deep hyper-3D convolutional network (3D-CNN) that combines 3D-CNN with 3D attention modules for white blood cell classification [14]. In addition to cell classification, deep learning has also been widely applied in medical image classification for cancer diagnosis [15]. Most relevant studies have used hyperspectral imaging and convolutional neural network (CNN) classifiers for cancer cell classification. For example, Sommer et al. used a CNN based on HSI data to classify nephrons, particularly residual neural networks (ResNets) [16]. Li et al. used ResNet34 deep learning architecture to classify gastric cancer on fluorescence hyperspectral images, which achieved over 96% in accuracy, specificity, and sensitivity in classification [17]. Bengs et al. used HSI and various deep learning methods in their more challenging in vivo tumor classification research [18]. Ma et al. combined wavelet transform features with machine learning and proposed a feature classification method based on the discrete wavelet transform (DWT) [19]. Collectively, these examples fully reflect the enormous potential of deep learning in the field of tissue or cell classification. With the continuous development of technology, we have reason to believe that deep learning will play a more important role in future in the field of medical diagnosis.

Although hyperspectral imaging technology has been widely applied and studied in other fields, there are few reports on its application in the rapid diagnosis of thyroid nodules during surgery, for which further exploration and research are needed. To address this, we propose a rapid method for the diagnosis of benign and malignant thyroid nodules based on hyperspectral technology. First, as the incidence rate of thyroid nodules in the population gradually increases, the difficulty of obtaining thyroid nodule samples decreases accordingly. To collect the hyperspectral data of thyroid nodules, we first developed a system for collecting thyroid tissue hyperspectral data, obtaining data for a large number of complete types of thyroid nodule samples, as the necessary foundation for subsequent research. Secondly, current research on medical hyperspectral image classification is mainly focused on pixel-based region segmentation, and there is still some distance from the practical application of tumor tissue benign and malignant classification in clinical practice [18]. We propose a binary classification method for distinguishing benign and malignant thyroid nodules based on hyperspectral data blocks, which provides strong support for the application of hyperspectral technology in the medical field. Finally, we investigate the impact of the size of hyperspectral data blocks on the classification of benign and malignant thyroid nodules. Based on the existing data features, we constructed a classification model for processing the hyperspectral three-dimensional data of thyroid nodules, further improving the accuracy and practicality of diagnosis. The results of this study not only fill the gaps in the field of thyroid nodule diagnosis, but also lay the foundation for the wider application of hyperspectral technology in the medical field, with the potential to achieve the rapid diagnosis of benign and malignant thyroid nodules during surgery.

## 2. Materials and Methods

### 2.1. Experimental Framework

The experimental framework of our study is shown in Figure 1. We collected hyperspectral thyroid nodule data using our self-developed system of equipment. All samples were labeled by professional surgeons, with information about detailed categories. During the data labeling process, two doctors, based on the clinical pathological diagnosis results, jointly marked areas as healthy or nodule in the collected data. These markings were cross-validated to finalize the labeling. Figure 2 shows the pseudo-color image subsequently synthesized from the collected hyperspectral images and the labeled image containing the labeled areas. In this work, the marked areas were classified by the surgeons to obtain clearly defined labels. Out of 150 cases, 80 were labeled with both malignant nodules and normal tissue regions, while 70 cases were labeled with both benign nodules and normal tissue regions. These data provide rich and accurate information for subsequent classification.

For the case dataset, we adopted the method of image cropping, to cut each case image into smaller hyperspectral data blocks [20]. We then constructed a deep learning algorithm classification model to extract the typical spectral features corresponding to nodule tissue. This model can classify thyroid nodule data blocks, toward realizing the rapid diagnosis of benign and malignant thyroid nodules. In order to optimize the classification performance, we compared the effects of different deep learning methods and different data block sizes on the classification results. Through experiments and data analysis, we selected the most suitable data cropping mode and classification algorithm for thyroid nodule classification. Finally, in order to apply the proposed deep learning model to automated intraoperative diagnosis in practice, we combined it with the previous processing flow to form a fast intraoperative benign and malignant classification diagnostic method.

### 2.2. Thyroid Tissue Hyperspectral Imaging System

In this study, a self-developed hyperspectral imaging system was used for data collection. The system structure is shown in Figure 3, mainly comprising a halogen lamp light source, built-in scanning spectral imaging system, control system, and data acquisition system. The original collection system, on which ours is based, had some inconveniences. For example, the inflexibility of device movement and the complexity of specimen movement and focal length determination affect, to some extent, the efficiency and accuracy of data collection. To address these issues, we designed a more comprehensive integrated system. In the new system, we added a movable specimen tray controlled by a stepper motor and an imaging lens holder controlled by a servo motor to achieve the precise adjustment of focal length. In addition, all these components are integrated into a portable shell (80 cm × 80 cm × 60 cm), greatly improving the portability and practicality of the system. The added two-dimensional electronic precision displacement table is used to hold tissues, with a range of 80 mm and a positioning accuracy of 2 μm. Its movement on a two-dimensional plane is controlled by transmitting information through a computer, which significantly improves the efficiency and accuracy of data collection of the new system, providing a more reliable and accurate data foundation for subsequent classification and diagnosis work. During the data collection process, the projected beam passes through the sliced sample and enters the scanning spectral imaging instrument (acA2000-165umNIR near-infrared enhanced sensor, designed by BASLER from Germany; wavelength coverage range: 400 nm~1000 nm). The light passes through a series of optical components, including slits, collimators, and dispersion elements, and finally shines onto the detector array. This not only produces one-dimensional spatial and spectral information but is also based on the utilization of halogen lamps (with a wavelength range of 400 nm to 2500 nm and a power of 50 W) as a stable lighting source to ensure the required illumination for the system. The data acquisition system stores images in real time, in addition to sending instructions to control the movement of the precision storage platform, thereby obtaining another dimension of spatial information for the target. Finally, the complete data are obtained. The collected data have a specific format: 700 (x) × 600 (y) × 294 (λ) × 12 (bit). Here, 700 × 600 represents the image size, 294 represents the number of spectral channels, and 12 represents the number of data bits per pixel in each band. In practical application, we only retain the visible and near-infrared spectral data of 555 nm–865 nm, excluding the initial and final 49 spectral bands. The data are stored in Band-Interleaved-by-Pixel (BIP) format, in pixel spectral order. In each case, the data can be stored in 5 s. The spectral resolution and spatial resolution of the system are 3 nm and 16.7 μm. In addition, the system also generates three-dimensional pseudo-color images that are easy to observe with the naked eye for annotating data categories.

### 2.3. Experinmental Datasets

During thyroid nodule resection surgery, we obtained images for preparing a hyperspectral image dataset for this work. All images were captured in the operating room using the acquisition system described in Section 2.2. Thus, we created a database that contains various images of thyroid tissue, including benign thyroid nodules, malignant thyroid nodules, and a small amount of normal thyroid tissue. Our complete dataset consisted of 150 patients and included hyperspectral data of thyroid nodule tissue. These patients underwent surgery due to thyroid nodules. These data were collected by the Thyroid Surgery Department of the First Affiliated Hospital of Zhengzhou University and collected using a hyperspectral imaging system. In the dataset, there were 80 patients of malignant thyroid nodules and 70 patients of benign thyroid nodules. The training set consisted of 105 patients, accounting for 70%, including 55 malignant nodules and 50 benign nodules.

During the data collection process, we also synthesized pseudo-color images representing the tissue region. These images were annotated by professional surgeons using the hospital pathology database. For each image, the annotated images were mainly divided into three classification groups: the first category was background, the second category was normal thyroid tissue, and the third category was benign or malignant thyroid nodules, as shown in Figure 2. In order to better meet the needs of the surgical process, we only used the extracted image data of the nodule area for classification, including benign and malignant thyroid nodules.

### 2.4. Methods

#### 2.4.1. Data Preprocessing

(1)Whiteboard correction and smoothing treatment

Before data collection, we performed critical calibration steps to ensure the accuracy and consistency of the data. We selected a white board with 50% reflectivity as the standard reference target for rigorously calibrating the hardware system of the microscopic hyperspectral equipment. This process was crucial for eliminating hardware differences between devices and ensuring data quality. Through such calibration, we can ensure that the data collected each time had a high degree of reliability and consistency.

The correction formula is as follows:$$S_{C}=\frac{S}{S_{W}-S_{D}}\times R_{W}$$
where $S_{C}$ is the corrected hyperspectral signal value of thyroid tissue, S is the collected hyperspectral signal value of thyroid tissue, $S_{W}$ is the hyperspectral signal value collected from the standard whiteboard, $S_{D}$ is the system dark signal value, and $R_{W}$ is the reflectance of a standard whiteboard. To obtain higher-quality hyperspectral data of thyroid nodules, the standard whiteboard reflectance used in this study was 50%. In order to obtain high-quality results, we used Savitzky–Golay filters to smooth the data [21], with a window length of 21, fitted with a fifth-order polynomial, and then normalized the data using a 0–1 normalization method.

(2)Data cropping

In order to increase the amount and diversity of data in addition to improving the stability of the model, we continuously segmented the preprocessed hyperspectral data in the spatial dimension through the selection of blocks of different sizes. In this way, the collected hyperspectral data were intercepted and randomly flipped, which was needed for processing the dataset. After these treatments, we ultimately obtained a large amount of benign hyperspectral data and malignant data. These data blocks differed in size, with dimensions of 20 × 20 × 196, 40 × 40 × 196, 50 × 50 × 196, and 80 × 80 × 196. The first and second dimensions are spatial dimensions, while the third dimension is the spectral dimension. The final number of data blocks that we obtained for analysis is shown in Table 1.

(3)Data dimensionality reduction

Compared to common 3D visible light images, hyperspectral images often contain subtle spectral differences, which is one of their advantages. However, this advantage also brings new challenges, with data processing becoming quite complex due to the inclusion of numerous bands. The correlation between bands leads to a large amount of information redundancy, especially when there is a high correlation between adjacent bands. In addition, their distribution in high-dimensional space causes the data to exhibit sparse and extremely irregular characteristics, making it difficult for traditional analysis algorithms to achieve ideal results. To address these issues, we chose to perform dimensionality reduction on the hyperspectral data. By reducing the dimensions, we can reduce the dimensionality of data while retaining its key features, thereby simplifying the process of data processing and analysis. Dimensionality reduction processing helps to reveal the intrinsic structure and patterns of data, improves the generalization ability of models, and enables us to better understand and utilize hyperspectral data.

We mainly used PCA to reduce the dimensionality of hyperspectral data blocks [22]. Multi-band images are converted into a new feature space through linear transformation, minimizing the covariance between samples in the new space, and preserving as much data information as possible while reducing data dimensions. We reduced the spectral dimension of hyperspectral images to 3 through PCA, thereby preserving the feature information of nodule data while greatly reducing computational complexity. 

#### 2.4.2. Model

(1)Feature extraction model

Hyperspectral data have three dimensions: spectrum, width, and height. In these three dimensions, the spatial information contained in the width and height dimensions is closely related to the spatial information in the spectral dimension.

In order to more effectively extract the spatial and spectral features of hyperspectral data of thyroid nodules, we used a 3D CNN model to extract the features of hyperspectral data. Compared to traditional 2D CNNs [23], there is an additional dimension in 3D CNNs [24], making it more suitable for processing 3D data, such as hyperspectral images [25]. The application of this model enables us to better analyze and understand the characteristics of hyperspectral data.

More specifically, a 3D CNN takes a continuous multi-channel input, which enhances the spectral dimension of information, resulting in the extraction of more rich and expressive features. 

In addition, the advantage of a 3D CNN lies in its ability to share convolution kernels, which allows network parameters to be reduced, thereby improving computational efficiency. When processing hyperspectral images, a 3D CNN can better utilize spatial and spectral information, thereby extracting richer features. 

(2)Feature processing and classification models

In Figure 4, we used a 3D CNN to extract high-dimensional features of thyroid nodule hyperspectral data. In order to effectively classify the features, we used Visual Geometry Group Network 16 (VGG16) to process and classify the extracted features [26]. Our VGG consisted of six block structures, each with the same number of channels. The first three block structures were composed of two convolutional layers, the fourth and fifth block structures were composed of three convolutional layers, and the sixth block structure was composed of three fully connected layers. Each block structure was connected by pooling layers, and the softmax function was used to complete the classification of benign and malignant thyroid nodules. All nonlinear elements used the ReLU function as the activation function.

We chose this architecture because it provides a simple and effective method for analyzing spectral information. Convolutional operations act on the local structure of the spectrum, and we used relatively small kernel sizes and stacked layers to increase the receptive field while improving computational efficiency [27]. Two fully connected layers made the final decision based on the global context. The advantage of this method is that it combines local and global information aggregation while still maintaining computational efficiency. Our model was named V3Dnet.

#### 2.4.3. Training and Evaluation

Due to the abundant information in the dataset, we cropped the data to enhance their diversity. We then split the cropped data into test and validation subsets. The validation subsets were subsequently used for tuning hyperparameters. We averaged the metrics of all tests for reports on testing performance. For data augmentation, we used random cropping and random flipping during training. To evaluate the performance of the labeled area, we used ordered crops and averaged the predicted values of all crops to obtain a classification. We trained the model for 300 iterations with a batch size of 32 and used the stochastic gradient descent optimizer for optimization. To address the issue of data imbalance, we added category weights when calculating classification losses, which are inversely proportional to the number of samples in the corresponding category.

After model training for each algorithm, the confusion matrix was immediately obtained after prediction validation or testing data. The confusion matrix included the four parameters of true positives (TP), true negatives (TN), false positives (FP), and false negatives (FN).

The overall accuracy index was calculated from the confusion matrix to select the best model. This was completed during cross-validation to ultimately classify the new image using the selected image. This indicator is described in the equation below, where all correct predictions are divided by all predictions made.

After predicting the entire patient image, four metrics were used to evaluate the performance of the selected model. These indicators were accuracy, sensitivity, precision, and *F1-score*.
$$ACC=\frac{TP+TN}{TP+TN+FP+FN}$$
$$SEN=\frac{TP}{TP+FN}$$
$$PRE=\frac{TP}{TP+FP}$$
$$F1-Score=2\times \frac{PRE\times SEN}{PRE+SEN}$$

This performance metric can be used to determine the degree to which the trained model classifies the data globally. On the other hand, SEN is used to determine the sensitivity of the classifier to the lesion area, while PRE indicates the accuracy of the model in predicting the lesion area. F1-score represents the harmonic mean of accuracy and recall, which comprehensively considers accuracy and recall. It is a comprehensive evaluation indicator and is commonly used as one of the important indicators of a model’s performance in classification.

## 3. Experimental Results

Due to the differences in the composition of biological substances that can be distinguished by their spectral characteristics, it is necessary to use large-scale hyperspectral datasets of thyroid tissue to analyze the spectral distribution and explore intrinsic connections. Statistical analysis and visualization on the spectral distribution of different tissues in all training sets are performed (Figure 5A). For ease of observation and comparison, lighter-colored areas represent the spectral span between the maximum and minimum values, visualized in the same color. As shown in Figure 5A, the hyperspectral curve characteristics of the same tissue are relatively similar. At the same time, there are differences in the amplitude of the increase and decrease in hyperspectral data of different categories of thyroid tissue with changes in spectral bands. To prove this point, we performed whiteboard correction on 10 benign cases and 10 malignant cases and sampled 150 spectral curves. After calculating their average values, we smoothed and differentiated the curves and obtained the results shown in Figure 5B, verifying the differences in changes in hyperspectral data of different tissues. Among them, the average curve of the raw sampled spectral data without processing at the sampling points is shown in the dark curve in Figure 5A.

Two experiments were conducted to evaluate the database and proposed methods for analyzing the optimal choice between benign and malignant thyroid nodules.

### 3.1. Model Results

To determine the effectiveness of V3Dnet, we conduct experiments using the hyperspectral data of thyroid nodules with a data block size of 50 × 50 × 196. Firstly, PCA is used to reduce the dimensionality of 196-dimensional hyperspectral data, changing the block size to 50 × 50 × 3. Then, two 3D CNNs are used to extract features from the hyperspectral data of thyroid nodules. The first 3D CNN has 8 convolution kernels, and the second 3D CNN has 32 convolution kernels. After passing through two 3D CNNs, the original data become 32 data blocks of size 50 × 50 × 3, which are then recombined to 50 × 50 × 96 in size, and PCA is then used to reduce the dimensionality of the data using the block size of 50 × 50 × 3. Through data preprocessing, mainly data color normalization and size correction, the previously processed data are transformed into data such that they can be input into VGG16, and then classified into benign and malignant. During the model training process, the training set accounts for 70% of the total number of samples (randomly selecting 1/7 as the validation set, equivalent to 10% of the total data), all from 105 training patients, and the test set accounts for 30%. In order to increase the generalization performance of the model, reduce the correlation between the training and testing sets, and improve the accuracy of clinical judgments, we generate training and testing sets from different cases during the dataset segmentation process. During each iteration of training, the training samples are shuffled again. The training batch is set to 64, the learning rate is set to 0.001, and a stochastic gradient descent optimizer is used. After 150 iterations, the training parameters are stable. The test set data are input into the model for testing, and the results are shown in Table 2. The test set contains a total of 1503 data points, 760 malignant nodules, and 743 for benign nodules. At the same time, we also compare models such as the 1D CNN [28], 2D CNN, 3D CNN, VGG16, and Spectral–Spatial Residual Network (SSRN) [29].

Based on the test results, we calculated the accuracy, precision, and specificity indicators to evaluate the performance of the model, as shown in Table 3. Our method achieved an accuracy of 84.63%, indicating that V3Dnet can effectively classify benign and malignant thyroid nodules.

Because the classification of malignant tissue is more important in clinical surgery, we analyzed the misdiagnosis rate of malignant nodules based on the classification results, as shown in Table 4. The results indicate that the misdiagnosis rate of V3DNET is relatively low.

As mentioned, we make the training and testing sets come from different cases as much as possible when dividing, which leads to a low correlation between the training and testing sets. At the same time, our testing set accounts for a large proportion, ensuring the generalization and classification performance of our model. In order to further increase the reliability of our results, we used a ten-fold cross-validation method for an unbiased estimation of prediction accuracy on a 50 × 50 × 196 dataset. The experimental results are shown in Table 5.

### 3.2. Results for Hyperspectral Data of Different Block Sizes

To determine the impact of rapid hyperspectral data block size on the classification performance of thyroid nodules, we compared the classification results of 20 × 20 × 196, 40 × 40 × 196, 50 × 50 × 196, and 80 × 80 × 196. As shown in Table 6, under the premise of convenient processing, the use of thyroid nodule hyperspectral data of a larger size was more helpful for the task of classification as benign or malignant.

Based on the test results, we calculated the accuracy, precision, and specificity indicators to evaluate the performance of the model, as shown in Table 7. The accuracy of our method reached 85.76% on the hyperspectral data test set of thyroid nodules with a block size of 80 × 80 × 196, indicating that increasing the block size in the spatial dimension affects the classification of benign and malignant thyroid nodules to a certain extent.

We compared V3Dnet with 3D CNN and SSRN for data cubes of different block sizes, as shown in Figure 6, where A shows the accuracy line chart for different block sizes and models, B shows the sensitivity line chart, C shows the precision line chart, and D shows the F1-score line chart.

## 4. Discussion

In this study, we compared and studied multiple network models in the classification of thyroid nodules based on their hyperspectral data. When attempting to establish a complex network model, we found that overfitting was quite severe, resulting in poor classification performance when using the test set. Therefore, we have decided to abandon overly complex network models and choose the smaller network VGG16 as the basic model, which has better classification performance and can fully extract features from the hyperspectral data. The two models demonstrated good performance in processing for the task of classifying thyroid nodules based on hyperspectral data, which was effectively completed. We tested the feature pyramid and 3DCCN method proposed by Chen et al. [30], mainly comparing the accuracy and testing time, and found that our method is simple, fast, and produces better results. The experimental parameters were set according to Section 3 and trained on a dataset for a block size of 50 × 50 × 196. The comparison results are shown in Table 8.

In addition, we attempted to use the complete hyperspectral data of thyroid nodules as a dataset for model training and testing. However, we found that the complete dataset contains too much interference from tissue edges and the background, as well as factors such as blood stains, which have a significant impact on the classification performance. In clinical application, it is not necessary to classify the entire image acquisition area. Due to the influence of the method of organizing regions proposed by Beng et al. [18], we ultimately chose to use a segmented data cube for our study. This can allow the better removal of interfering factors, as well as an improved accuracy and reliability of classification, and this approach is more amenable to clinical application.

In future research, we will further optimize the network model and data processing methods to improve the accuracy and reliability of thyroid nodule classification based on hyperspectral data. We are preparing to compare and draw on the dynamic residual method proposed by Wang et al. [31] and the Gabor wavelet transform feature extraction method proposed by Huang et al. [32]. These are two completely different approaches that may be helpful for our work. This will aid in promoting the application of this technology in clinical diagnosis and treatment, providing doctors with more accurate and reliable diagnostic evidence, and thus better serving patients.

We conducted in-depth experiments and analysis to study the influence of model size on the performance in thyroid nodule data classification. The results show that as the size of the data block gradually increases, the classification performance shows a gradually improving trend. This may be because data with blocks of a larger size can provide more information, enabling the model to better learn and recognize the features of thyroid nodules.

However, when the size of the data blocks is increased to a certain point, the number of blocks that can be cropped will sharply decrease due to the size limitations of thyroid nodules themselves and issues during the data cube cropping process. This may lead to a significant decrease in the sample size that the model can use during training, thereby affecting further improvement in classification performance.

Therefore, we suggest using data blocks of size 50 × 50 for subsequent experiments. This size should be based on the size of thyroid nodules, taking into account both data size and volume, providing sufficient samples for model training and testing, and thus completing the task of thyroid nodule classification as benign and malignant. By making such a choice, we can ensure effective classification while avoiding the problem of insufficient sample size caused by the large size of the data segment.

## 5. Conclusions

In our study, we aimed to develop a novel classification method for benign and malignant thyroid nodules, which has clinical value for the application of hyperspectral data cubes. The existing classification methods are mainly based on hyperspectral data of pixels, and we propose a new method using hyperspectral data blocks. Using this type of data cube is more practical in clinical diagnosis and they can provide more tissue information. For this new method, we have developed a preprocessing and classification model for this dataset for the classification of benign and malignant thyroid nodules. To validate our method, we established a dataset for this classification, including 70 cases of benign nodules and 80 cases of malignant nodules. We also created a corresponding dataset to better utilize hyperspectral information. In the experiment, we tested on a sample of block size 80 × 80 × 196 and achieved an accuracy of 84.63%. This result indicates that our method has high classification accuracy and can be effectively used for the classification of benign and malignant thyroid nodules.

In addition to the classification effect, the thyroid nodule collection system we have established also has important value. This system can not only be used for classification tasks, but also provide more comprehensive organizational information for pathological diagnosis. In the next stage, we plan to further expand the dataset and collect more rare and segmented cases of thyroid nodules to improve the accuracy of classification. After improving the classification accuracy, we will conduct in vivo classification experiments during thyroid nodule surgery to reduce harm to patients and improve the safety and accuracy of surgery.

## Figures and Tables

**Figure 1 sensors-24-03197-f001:**
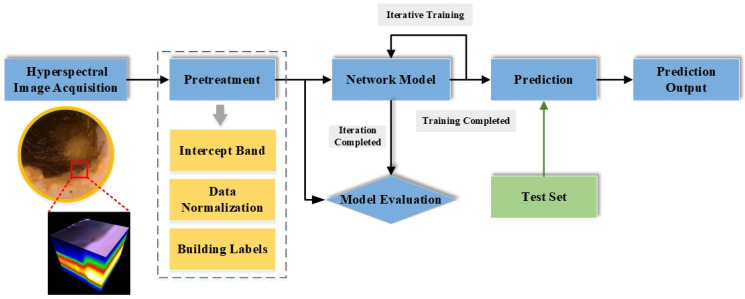
Experimental framework.

**Figure 2 sensors-24-03197-f002:**
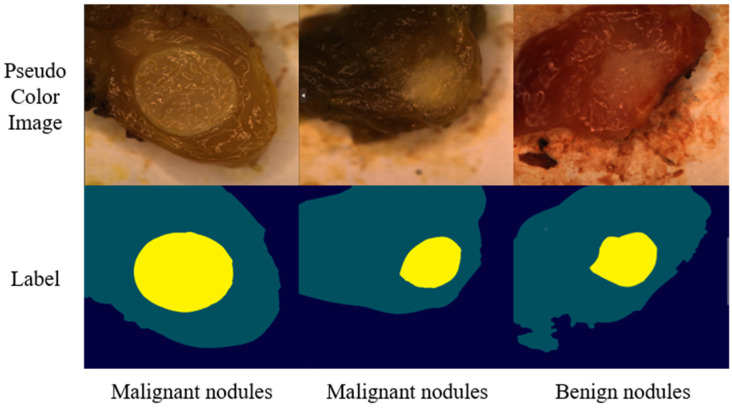
Pseudo-color images and labels of thyroid nodules.

**Figure 3 sensors-24-03197-f003:**
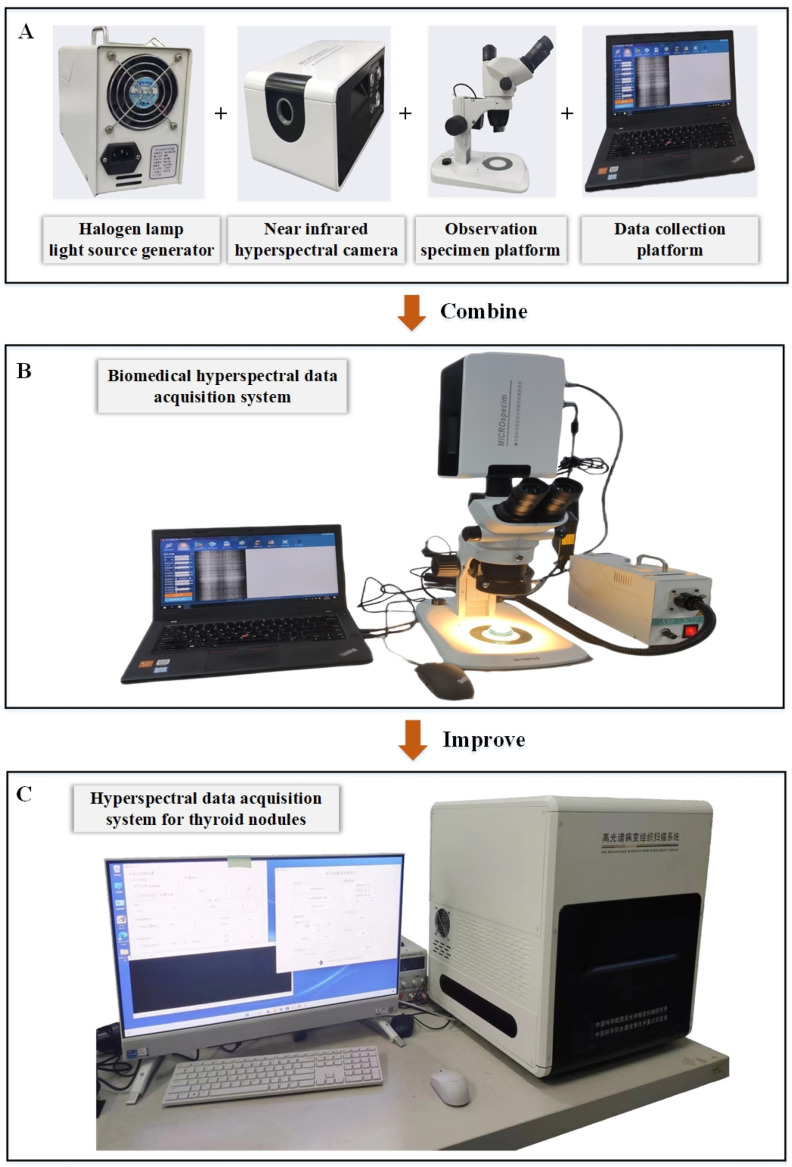
Thyroid tissue hyperspectral imaging system. (**A**) System components. (**B**) Original collection system. (**C**) Improved collection system.

**Figure 4 sensors-24-03197-f004:**
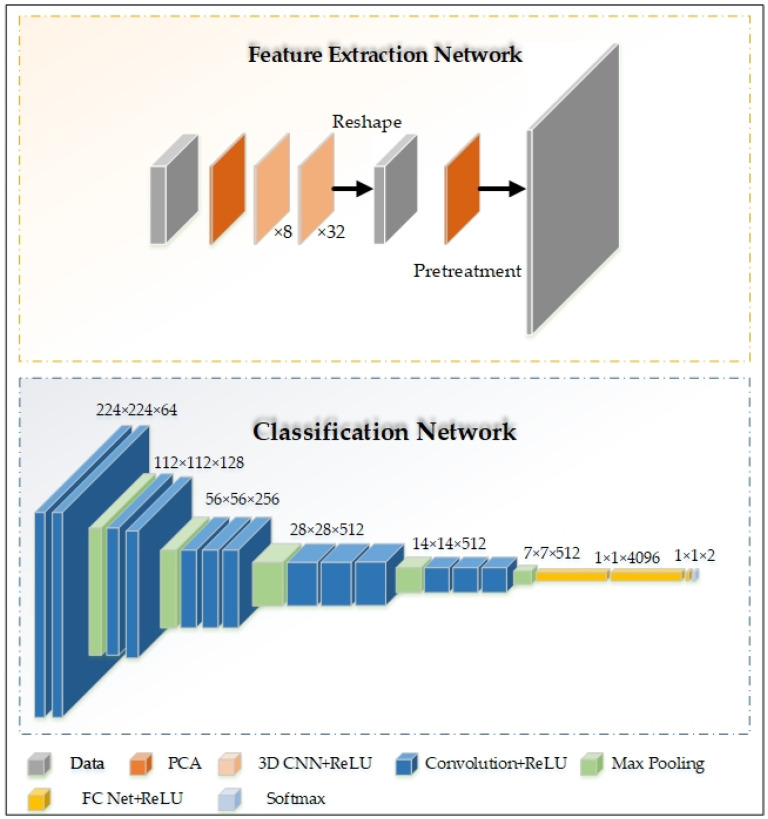
Model structure.

**Figure 5 sensors-24-03197-f005:**
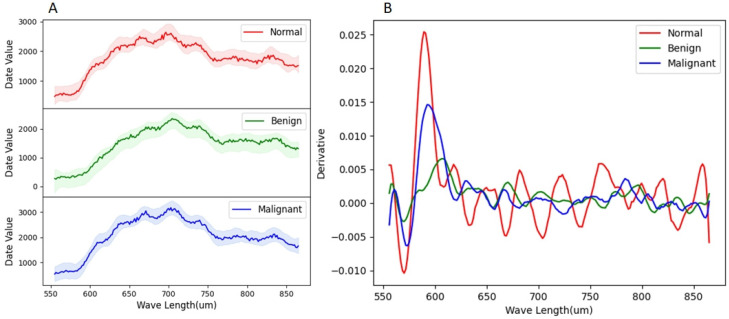
(**A**) Spectral profiles of three organizational categories. The light-colored area represents the spectral span between the maximum and minimum values of the statistics, while the dark center curve represents the average spectral distribution of the sampling points. (**B**) Differential graph of average spectral curves at sampling points of different organizations.

**Figure 6 sensors-24-03197-f006:**
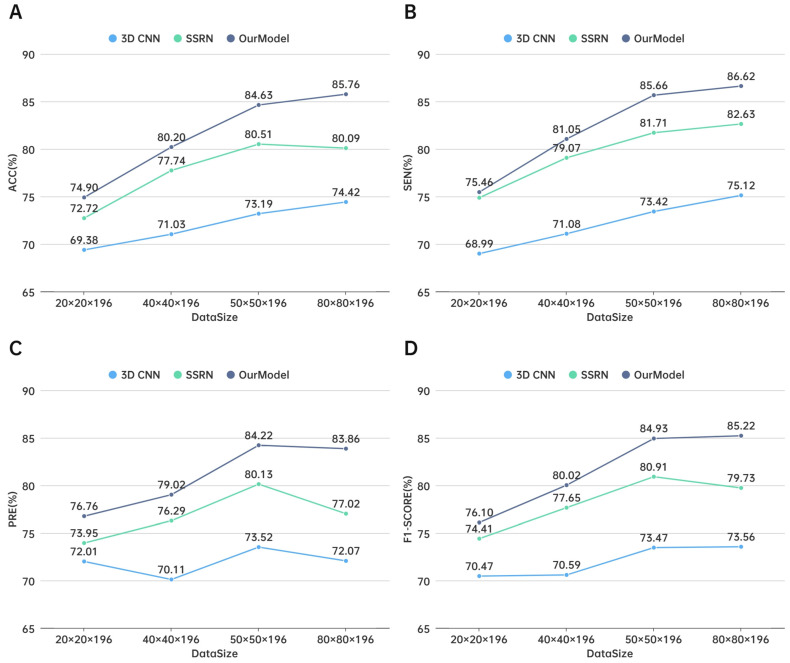
Line chart results for different block sizes and models: (**A**) accuracy; (**B**) sensitivity; (**C**) precision; (**D**) F1-score.

**Table 1 sensors-24-03197-t001:** Experimental dataset.

Block Size	Malignant Nodule Data Blocks	Benign Nodule Data Blocks	Total
20 × 20 × 196	4538	4029	8567
40 × 40 × 196	3375	3527	6902
50 × 50 × 196	2534	2477	5011
80 × 80 × 196	1419	1578	2997

**Table 2 sensors-24-03197-t002:** Test results for each model.

Model	TP	FN	FP	TN
1D CNN	547	213	215	528
2D CNN	555	205	194	549
3D CNN	558	202	201	542
VGG16	594	166	171	572
SSRN	621	139	154	589
V3Dnet	651	109	122	621

**Table 3 sensors-24-03197-t003:** Classification results for each model.

Model	ACC	SEN	PRE	F1−Score
1D CNN	71.52	71.97	71.78	71.87
2D CNN	73.45	73.03	74.10	73.56
3D CNN	73.19	73.42	73.52	73.47
VGG16	77.58	78.16	77.65	77.90
SSRN	80.51	81.71	80.13	80.91
V3Dnet	84.63	85.66	84.22	84.93

**Table 4 sensors-24-03197-t004:** Type I error for each model.

Model	1D CNN	2D CNN	3D CNN	VGG	SSRN	V3Dnet
Type I Error (%)	28.93	26.11	23.01	77.32	20.72	16.42

**Table 5 sensors-24-03197-t005:** Classification results for ten-fold cross-validation.

Model	1D CNN	2D CNN	3D CNN	VGG	SSRN	V3Dnet
ACC	70.94	72.89	72.93	78.32	80.15	84.46

**Table 6 sensors-24-03197-t006:** Test results for each block size.

Size	TP	FN	FP	TN	Total
20 × 20 × 196	1027	334	311	898	2570
40 × 40 × 196	821	192	218	840	2071
50 × 50 × 196	651	109	122	621	1503
80 × 80 × 196	369	57	71	402	899

**Table 7 sensors-24-03197-t007:** Classification results of hyperspectral data blocks of different sizes.

Size	ACC	SEN	PRE	F1−Score
20 × 20 × 196	72.53	73.03	74.57	73.79
40 × 40 × 196	76.24	77.00	75.07	76.02
50 × 50 × 196	84.63	85.66	84.22	84.93
80 × 80 × 196	85.76	86.62	83.86	85.22

**Table 8 sensors-24-03197-t008:** Comparison of accuracy and training time.

Model	ACC	Training Time (min)
FPN + 3D CNN	72.53	58
V3Dnet	80.64	34

## Data Availability

The data that support the findings of this study are openly available in zenodo. DOI: 10.5281/zenodo.11180146.

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
