# Peer review of "Classification of Benign–Malignant Thyroid Nodules Based on Hyperspectral Technology"

_sensors, 2024, doi:10.3390/s24103197_

Round 1

Reviewer 1 Report

Comments and Suggestions for Authors

Review

The manuscript „Classification of benign-malignant thyroid nodules based on hyperspectral technology“ presents results of a thorough experimental work, which could be valuable. The authors assembled a setup for obtaining hyperspectral images of thyroid samples and made measurements on extensive experimental material. However, the proposed classification method leaves many questions. First of all, it would be necessary to show the possibilities of simpler, well-established methods of discrimination. In particular, it was necessary to try linear chemometrics algorithms on unfolded hyperspectral data in order to see the internal data structure using score plots. This would allow the estimate the separation potential of different tissues to be assessed. Discriminant analysis using the PLS-DA method would also be useful to evaluate the performance of the method. Only after this, the use and comparative study of nonlinear methods such as neural networks is justified.

The methods used are not fully described and are not always justified. Spectral data is not shown in a way that allows visual assessment of spectral differences compared to measurement reproducibility. The quality of English is formally good, but the peculiar style of presentation is difficult to understand. It could be shorter, but more informative, instead.

Therefore, although the work has potentials for publication, I am forced to reject it due to serious errors and the poor technical quality of the manuscript.

Please find some specific comments below.

Explain the choice of wavelength region 555-865 nm. Why the main NIR interval was excluded, although it may contain useful information?

Lines 231-234 and 240-243 contain the same information.

Lines 260-261: Provide full details on Savitzky-Golay smoothing (window, polynomial order) and subsequent normalization algorithms.

Section 2.4.1

2) Data interception: It is not clear, how the data truncation is used to increase the amount of data cubes. In Table 1 indicate clearly, which dimensions are spatial and which one contains spectral variables.

Line 271: “Hyperspectral data can reveal subtle differences in spectral characteristics, which is its core advantage.” It is unclear what advantage the authors mean, which is not related to the spectroscopy in general.

“crucial, crucial, extreme, core“ etc. – too much unnecessary emotional adjectives

Line 293: This normalization algorithm is usually called “autoscaling” in chemometrics

3) Data dimensionality reduction

This sub-section describes PCA method. There is no need to provide details for this journal. It would be enough to provide a standard reference. Also, Fig. 4 is not necessary as an example. PC-images should belong to the Results and discussion part.

Line 336: 3D CNN can better utilize spatial and temporal information” - There is no temporal information in the data.

Abbreviations like CNN and VGG should be introduced at the first mentioning.

Due to the abundant information in the dataset, we cropped the data to enhance the 356 diversity of the urban area. What is meant by “urban area”?

...and Adam optimized it. What does it mean?

To address category imbalance, we inversely proportional the losses of each category to the samples of each category.“ It is not clear.

A lot of redundancy and unnecessary repetitions in the text, for example:

This performance metric can be used to determine the degree to which the trained model classifies the data globally. On the other hand, SEN is used to determine the sensitivity of the classifier to the lesion area, while PRE indicates the accuracy of the model's 385 prediction of the lesion area. F1 Score represents the harmonic mean of accuracy and recall, which comprehensively considers accuracy and recall. It is a comprehensive evaluation indicator and is commonly used as one of the important indicators of classification 388 model performance.

Results section

Figure 6. Original spectra (before taking the derivative) should be also shown in the figure. What derivative was used? First, second? Which algorithm and options were used to calculate it? Normal tissue should be shown besides the benign and malignant samples. X-axis should be the spectral scale in nm.

Line 409-410: “PCA is used to reduce the dimensionality of 196-dimensional hyperspectral data, making the data cube the size of 50×50×3.“ - The choice of (only?) 3 PCs in the model should be explained. When PCA is used for data compression, it is important not to lose relevant information in rejected components.

Line 427-428: “Our method achieved an accuracy of 84.63%, indicating that our model can effectively classify benign and ma lignant thyroid nodules.“ - 85% is a poor classification accuracy for cancer diagnostics, also consider the number of false negatives.

There is no need to call the hyperspectral image data “cubes”. It is always 3D data, but not necessarily a cube.

Comments on the Quality of English Language

English formal quality is good, but the style is inappropriate for a scientific paper

Author Response

1.Explain the choice of wavelength region 555-865 nm. Why the main NIR interval was excluded, although it may contain useful information?

Lines 231-234 and 240-243 contain the same information.

Lines 260-261: Provide full details on Savitzky-Golay smoothing (window, polynomial order) and subsequent normalization algorithms.

Response:

We greatly appreciate the valuable feedback from the reviewers.

(1) When designing the data acquisition system, we compared the wavelength range of 400nm-1200nm and found that hyperspectral images with longer wavelengths have lower spatial resolution for biological tissues. So we ultimately chose a hyperspectral detector with a wavelength range of 400nm-1000nm to form a data acquisition system. Due to the influence of the detector's spectral response function, the data we collected had the best quality in the wavelength range of 555nm-865nm, so we chose the hyperspectral data of 555-865nm.

(2) The same information appeared because it was accidentally touched during literature processing and has been edited and modified. We have made detailed modifications to the entire text and provided assistance with MDPI English editing.

(3)  Through testing and comparison, when using the Savitzky-Golay filter, we chose a 5th order polynomial fitting with a window length of 21. After that, we used the 0-1 normalization method to normalize the data to the 0-1 range for data processing.

2.Section 2.4.1

2) Data interception: It is not clear, how the data truncation is used to increase the amount of data cubes. In Table 1 indicate clearly, which dimensions are spatial and which one contains spectral variables.

Line 271: “Hyperspectral data can reveal subtle differences in spectral characteristics, which is its core advantage.” It is unclear what advantage the authors mean, which is not related to the spectroscopy in general.

 “crucial, crucial, extreme, core“ etc. – too much unnecessary emotional adjectives

Line 293: This normalization algorithm is usually called “autoscaling” in chemometrics

Response:

We greatly appreciate the reviewer's careful reading of the manuscript and pointing out these issues.

(1) We continuously segment the preprocessed hyperspectral data in the spatial dimension by selecting boxes of different sizes, with the first and second dimensions being the spatial dimension and the third dimension being the spectral dimension. Therefore, we extract the collected hyperspectral data and use it as the processing dataset. The advantage of hyperspectral data is that it increases spectral details compared to RGB images.

(2) We have made comprehensive revisions to the language issues raised by the reviewers and look forward to your correction and comments.

3.3) Data dimensionality reduction

This sub-section describes PCA method. There is no need to provide details for this journal. It would be enough to provide a standard reference. Also, Fig. 4 is not necessary as an example. PC-images should belong to the Results and discussion part.

Line 336: 3D CNN can better utilize spatial and temporal information” - There is no temporal information in the data.

Abbreviations like CNN and VGG should be introduced at the first mentioning.

Due to the abundant information in the dataset, we cropped the data to enhance the 356 diversity of the urban area. What is meant by “urban area”?

...and Adam optimized it. What does it mean?

„To address category imbalance, we inversely proportional the losses of each category to the samples of each category.“ It is not clear.

A lot of redundancy and unnecessary repetitions in the text, for example:

“This performance metric can be used to determine the degree to which the trained model classifies the data globally. On the other hand, SEN is used to determine the sensitivity of the classifier to the lesion area, while PRE indicates the accuracy of the model's 385 prediction of the lesion area. F1 Score represents the harmonic mean of accuracy and recall, which comprehensively considers accuracy and recall. It is a comprehensive evaluation indicator and is commonly used as one of the important indicators of classification 388 model performance.“

Response:

We greatly appreciate the constructive feedback provided by the reviewer.

(1) We have made deletions and modifications to address the content redundancy raised by the reviewers, making the article more concise and clear.

(2) We have made revisions to the relevant document editing errors to make the article clearer and more understandable.

(3) We have made modifications to the model and method citation process throughout the entire text to make its logic clear.

(4) We added category weights when calculating the classification loss, which are inversely proportional to the number of samples in the corresponding category.

Thank you again for the suggestions and assistance provided by the reviewer.

4.Results section

Figure 6. Original spectra (before taking the derivative) should be also shown in the figure. What derivative was used? First, second? Which algorithm and options were used to calculate it? Normal tissue should be shown besides the benign and malignant samples. X-axis should be the spectral scale in nm.

Line 409-410: “PCA is used to reduce the dimensionality of 196-dimensional hyperspectral data, making the data cube the size of 50×50×3.“ - The choice of (only?) 3 PCs in the model should be explained. When PCA is used for data compression, it is important not to lose relevant information in rejected components.

Line 427-428: “Our method achieved an accuracy of 84.63%, indicating that our model can effectively classify benign and ma lignant thyroid nodules.“ - 85% is a poor classification accuracy for cancer diagnostics, also consider the number of false negatives.

There is no need to call the hyperspectral image data “cubes”. It is always 3D data, but not necessarily a cube.

Response:

We greatly appreciate the questions raised by the reviewer.

(1) Based on the reviewer's suggestion, we have redrawn the image and first added the original hyperspectral image of the unprocessed thyroid nodule, as shown in Figure 6 (A). And 10 cases of benign nodules and 10 cases of malignant nodules were re selected. 50 spectral curves were sampled from benign nodule tissue, malignant nodule tissue, and normal tissue, respectively. After calculating their average values, smoothing and differentiation were performed to obtain the results shown in Figure 6 (B).,

(2) When processing hyperspectral data, in order to reduce data redundancy, dimension selection processing is usually performed on hyperspectral data. During the pre-experiment, we compared the selected 3 bands, 30 bands, 24 bands, and 64 bands for classification. We found that selecting too many bands did not have a positive impact on the classification results, but the model learning time was greatly increased. In order to improve the efficiency of model training and meet clinical needs, we ultimately chose to use PCA dimensionality reduction method to select the first three principal components for training.

(3) A classification accuracy of around 85% may not be the gold standard for classifying thyroid nodules, but it is already considered an excellent result among current relevant methods. At the same time, our goal is to make a preliminary classification of benign and malignant thyroid nodules within a limited time during surgery, which is conducive to guiding the operation. Currently, it has certain practical significance. Meanwhile, as the number of datasets increases, the accuracy of our proposed new method will further improve, becoming a more convincing method in clinical practice.

Reviewer 2 Report

Comments and Suggestions for Authors

This is a very interesting paper about the diagnosis of thyroid cancer using AI-enhanced hyperspectral imaging. However I have two significant concerns.

1) How original is this paper? I found another published paper from different authors, which is very similar to the one I am reviewing. This paper is  

Xi, N.M., Wang, L. & Yang, C. Improving the diagnosis of thyroid cancer by machine learning and clinical data. Sci Rep 12, 11143 (2022). https://doi.org/10.1038/s41598-022-15342-z

And it is not included into the list of references of the paper by Wang et al. The authors must include a review of the above paper into their introduction/discussion.  

2) The paper lacks statistical analysis. Of the total 150 datasets 100 were used for the deep learning training, and only 50 were used for the method testing. Is this sample size sufficient? What is the probability of type I error? The authors should perform this analysis.

Author Response

1.How original is this paper? I found another published paper from different authors, which is very similar to the one I am reviewing. This paper is  

Xi, N.M., Wang, L. & Yang, C. Improving the diagnosis of thyroid cancer by machine learning and clinical data. Sci Rep 12, 11143 (2022). https://doi.org/10.1038/s41598-022-15342-z

And it is not included into the list of references of the paper by Wang et al. The authors must include a review of the above paper into their introduction/discussion. 

Response:

We greatly appreciate the valuable opinions and suggestions of the reviewers. This study aims to use hyperspectral technology to propose a system that can collect thyroid nodule data during surgery and use deep learning algorithms to recognize and classify the data. The system can provide timely guidance for surgery based on the classification effect. We respond to the clinical demand for rapid classification of thyroid nodules during surgery by selecting hyperspectral methods that can carry more information. We collect data from freshly excised original thyroid tissue and use deep learning to quickly analyze thyroid nodules, providing timely guidance for ongoing surgery. At the same time, we have carefully studied the recommended article and found that its ideas and methods in the diagnosis of thyroid cancer are very informative. It has been added to the reference list. And its research methods and processes provide some guidance for our future research.

2.The paper lacks statistical analysis. Of the total 150 datasets 100 were used for the deep learning training, and only 50 were used for the method testing. Is this sample size sufficient? What is the probability of type I error? The authors should perform this analysis.

Response:

Thank you very much for the questions raised by the reviewer. Firstly, we utilized hyperspectral technology to enrich the spectral information of the collected data as much as possible, which is beneficial for our data processing. Meanwhile, we utilized data from different cases to crop their different regions, increasing their data diversity. When segmenting the training and testing sets, we strive to ensure that they originate from different pathologies, in order to reduce the correlation between the training and testing sets and improve the generalization performance of the model. At present, a good classification of benign and malignant thyroid nodules has been achieved, which can guide the progress of surgery to a certain extent. However, further classification of specific types of nodules cannot be achieved, and further data collection is needed to achieve the classification effect of specific types of nodules. Meanwhile, because the classification of malignant nodules is more important in the classification process, we analyzed the misdiagnosis rate based on the classification results of benign and malignant thyroid nodules. The analysis results are shown in Table 4. Our method has made certain progress in misdiagnosis rate compared to other common methods.

Reviewer 3 Report

Comments and Suggestions for Authors

Authors developed a thyroid tissue hyperspectral data collection system and proposed a deep learning approach to a binary classification task of thyroid nodules: benign or malignant.

The authors develop a relevant work in an important area. Nevertheless the paper requires some re-writing and clarification. The most critical and important point to clarify is the data pipeline transformations and division (into train-test and validation?) and the train, hyperparameter-tuning, and test procedures. Since this is mandatory to avoid biased test results.

As a minor issue I suggest that some short name should be applied to the proposed algorithm so that it is not referred to as “Our Model”.

Below there are a collection of comments that should be addressed by the authors:

l. 130 - “We propose a benign and malignant classification method based”

is it a binary classification problem? this should be clear right from the beginning of the paper, so please state clearly the task in this paper as a “binary classification with classes benign and malignant” to avoid confusion with other classification e.g. as Bethesda System for Reporting referred in section 1/Introduction.

l. 146. - Describe the labeling process. Each sample was labeled by one doctor ? Or did you use a more complex system ? (e.g. several doctors?, consensus?, etc …)

l. 156 and after. - pipeline

- data cropping into smaller blocks [25]

- extract spectral features

- deep learning model

shallow description

l. 176 - Hyperspectral Imaging System

- halogen lamp light; control system; data acquisition system; movable specimen tray controlled by a stepper motor; portable shell, etc. A more detailed description of the hardware including diagrams would be fully valorized in the scope of this Journal.

l. 209 – 2.3 Experinmental Datasets

“Experinmental” correct to “Experimental”

Furthermore, in section 2.3 the description of the procedures over the datasets is not clear enough.

Considering that from one side these include a division according to:

- background,

- the second category is normal thyroid tissue,

- third category is benign or malignant thyroid nodules,

is also mentioned the division among….

- papillary thyroid carcinoma,

- nodular thyroiditis

- normal thyroid nodule tissue

also the 150 cases divided into:

- 80 cases of data cube for papillary thyroid carcinoma and

- 70 cases of data cube for nodular thyroiditis.

and the training (2/3)

- 50 benign + 50 malignant

and test (1/3)

- 20 benign + 30 malignant

So the data pipeline must be explained in a more clear way, and when the authors are making comments about the data, it should be clear what point of the pipeline the comments are referring to.

(in section 3 the training/ division is also mentioned)

l. 262 - what is the meaning of this section title ? (since this seems to be a "image tiling" or "image patching." procedure)

l.270 - How to justify the number of data cubes presented table 1 ? these are a consequence of the block/patch dimensions but not in a direct way. If the data pipeline is explained in a more clear way this should be clarified.

l. 285 to 315 – I do not consider the explanation of standard PCA, citation to standard reference with PCA method should be enough. But indicating the number of reduced dimensions and the eigenvalues that justify this choice should be presented.

l. 357 - “we cropped the data to enhance the diversity of the urban area” it seems this sentence refers to some remote sensing application.

With respect to feature extraction/models a clear definition of the applied models should be presented, with a clear definition of its structure (layers, kernel dimensions etc) or a citation to a publication that presents the model in a detailed way.

These refer to 1D CNN, 2D CNN, 3D CNN, but also to VGG.

Which VGG? VGG16 is referred only in line 458 please specify as soon as VGG is referred in text that VGG is VGG16, if that is the case. Include a citation to VGG16 preferably to the original paper the proposes VGG16 (Simonyan, 2014) unless a variant is being considered.

l. 421 - “random gradient descent” is it stochastic gradient descent ?

l. 425 refers to SSRN – What is SSRN?

l. 460 – “these two models” which models refers to ? VGG16 and what else?

l. 462 - “We tested the feature pyramid and 3DCCN method proposed by Chen et al. [37]. Our method is simple, fast, and has better performance.” Where are these test results ? If they are being compared it should be clear what is being compared.

In section 3 it must be clarified how the data set is divided into train/test or train/validation/test and with what percentages. It is mentioned that:

- 70% of the total number of samples are used as the training set,

- 30% are used as the test set with 1503 data points (is this 30% of 5011? for 50x50 blocks, please clarify since there are 4 different blocks in table 1)

Furthermore, there is no reference to the validation set (is validation a part of the training set ? What percentage ? ) and the total amount of In fact the hyper-parameter tuning is mentioned in line 358 but not in the experimental results section. Without a clear procedure of train, hyper-parameter tuning, and test, with clearly defined datasets we may face biased performance results.

Table 2 and 3 refer to 50x50 blocks?

Table 3 and 4 How are these tests sets?

Figure 7 are not histograms. These are bar graphs and they do not add any value to the work. Well organized tables present the same results in an efficient way.

The discussion and conclusion sections should be “4.” and “4)” ?

The suggested size of data blocks is “50×50 for subsequent experiments”, but the performance (81.87% accuracy) is presented for 80x80 blocks. These two aspects should be coherent and present the performance of the suggested block.

Comments on the Quality of English Language

Only a minor typo was detected: “Experinmental” and a probable translation confusion between "random" and "stochastic"

Author Response

1. l.130 - “We propose a benign and malignant classification method based”

is it a binary classification problem? this should be clear right from the beginning of the paper, so please state clearly the task in this paper as a “binary classification with classes benign and malignant” to avoid confusion with other classification e.g. as Bethesda System for Reporting referred in section 1/Introduction.

Response:

Thank you very much for the valuable suggestions from the reviewer. This is a binary classification problem for benign and malignant classification, and the entire content has been modified to make it more detailed and understandable.

2. l. 146. - Describe the labeling process. Each sample was labeled by one doctor ? Or did you use a more complex system ? (e.g. several doctors?, consensus?, etc …)

Response:

Thank you very much for the reviewer's question. We used two experienced doctors to annotate the hyperspectral thyroid data we collected, including region segmentation and category labeling, cross validation, and finally completed the labeling.

3.l. 156 and after. - pipeline

- data cropping into smaller blocks [25]

- extract spectral features

- deep learning model

shallow description

Response:

Thank you very much for the suggestions put forward by the reviewer. We have made efforts to delete and modify this section of the description to ensure that it is concise and clear, and can clearly explain the method process.

4. l. 176 - Hyperspectral Imaging System

- halogen lamp light; control system; data acquisition system; movable specimen tray controlled by a stepper motor; portable shell, etc. A more detailed description of the hardware including diagrams would be fully valorized in the scope of this Journal.

Response:

Thank you very much for the reviewer's suggestions. The imaging system we ultimately improved and developed uses a two-dimensional electronic precision displacement table to hold tissue samples, with a range of 80mm and a positioning accuracy of 2 μ m. Control its movement through computer transmission of control signals. The acA2000-165umNIR near-infrared enhanced sensor (wavelength coverage range: 400nm-1000nm) was used, and a halogen lamp (wavelength range: 400nm-2500nm, power 50W) was used as a stable lighting source to ensure the required illumination for the system. Its spectral resolution and spatial resolution are 3nm and 16.7, respectively μ m. Finally integrated into a square shell of 80cm x 80cm x 60cm. The relevant content has been integrated into section 2.2 of the article.

5. l.209 – 2.3 Experinmental Datasets

“Experinmental” correct to “Experimental”

Furthermore, in section 2.3 the description of the procedures over the datasets is not clear enough.

Considering that from one side these include a division according to:

- background,

- the second category is normal thyroid tissue,

- third category is benign or malignant thyroid nodules,

is also mentioned the division among….

- papillary thyroid carcinoma,

- nodular thyroiditis

- normal thyroid nodule tissue

also the 150 cases divided into:

- 80 cases of data cube for papillary thyroid carcinoma and

- 70 cases of data cube for nodular thyroiditis.

and the training (2/3)

- 50 benign + 50 malignant

and test (1/3)

- 20 benign + 30 malignant

So the data pipeline must be explained in a more clear way, and when the authors are making comments about the data, it should be clear what point of the pipeline the comments are referring to.

(in section 3 the training/ division is also mentioned)

Response:

Thank you very much for the constructive feedback from the reviewer. In the previous description, we wanted to express our classification ideas as much as possible, but there were errors in the actual description. We differentiate thyroid nodule data into benign nodules and malignant nodules, and currently, this study only assesses their benign and malignant characteristics. When distinguishing between the training and testing sets, in order to improve the effectiveness of the model, we strive to balance the number of benign and malignant nodules in the training set as much as possible, and ensure that the training and testing sets come from different cases as much as possible. We have revised the description in Section 2.3, and the distinction between the actual training and testing sets is carried out after dividing them into data blocks. We have made efforts to revise the data differentiation approach throughout the article to make the description clearer. At the same time, we have screened and revised the description throughout the entire article, and have sought help from the official English editing service of MDPI to make the article more coherent.

6. l. 262 - what is the meaning of this section title ? (since this seems to be a "image tiling" or "image patching." procedure)

l.270 - How to justify the number of data cubes presented table 1 ? these are a consequence of the block/patch dimensions but not in a direct way. If the data pipeline is explained in a more clear way this should be clarified.

l. 285 to 315 – I do not consider the explanation of standard PCA, citation to standard reference with PCA method should be enough. But indicating the number of reduced dimensions and the eigenvalues that justify this choice should be presented.

Response:

Thank you very much for the reviewer's question.

(1) We changed the title to data cropping and provided detailed descriptions during the operation. We continuously segmented the preprocessed hyperspectral data in the spatial dimension by using selection boxes of different sizes.

(2) Due to the different characteristics of nodules in different regions, we chose to crop the data in order to improve the effectiveness of the model, thereby increasing data diversity. We also discussed the impact of data block size on classification results. At the same time, the collected individual case data has a lot of interference background. We have chosen to segment the nodule area, which is beneficial for reducing background interference in the data and solving the classification requirements for nodule areas in clinical applications more quickly, accurately, and conveniently.

(3) We have made modifications to the citation of PCA in the text. Usually, when processing hyperspectral data, in order to reduce data redundancy, dimension selection processing is usually performed on hyperspectral data. During the pre experiment, we compared the selected 3 bands, 30 bands, 24 bands, and 64 bands for classification. We found that selecting too many bands did not have a positive impact on the classification results, but the model learning time greatly increased. Therefore, we ultimately chose to use PCA dimensionality reduction method to train the first three principal components of spectral feature selection.

7. l. 357 - “we cropped the data to enhance the diversity of the urban area” it seems this sentence refers to some remote sensing application.

With respect to feature extraction/models a clear definition of the applied models should be presented, with a clear definition of its structure (layers, kernel dimensions etc) or a citation to a publication that presents the model in a detailed way.

These refer to 1D CNN, 2D CNN, 3D CNN, but also to VGG.

Which VGG? VGG16 is referred only in line 458 please specify as soon as VGG is referred in text that VGG is VGG16, if that is the case. Include a citation to VGG16 preferably to the original paper the proposes VGG16 (Simonyan, 2014) unless a variant is being considered.

l. 421 - “random gradient descent” is it stochastic gradient descent ?

l. 425 refers to SSRN – What is SSRN?

l. 460 – “these two models” which models refers to ? VGG16 and what else?

Response:

Thank you very much for the reviewer's suggestions and suggestions. We have removed editing errors such as "urban area". The model is our proposed V3Dnet. In response to editing errors, we have made comprehensive revisions to the article to make the explanations more understandable. We have also sought help from MDPI's official English editing service and made extensive revisions to the entire text. "Random gradient descent" is "stochastic gradient descent", and SSRN is "Spectral-Spatial Residual Network". We have modified the citation process of all models and methods in this article to make their logic clear.

8. l. 462 - “We tested the feature pyramid and 3DCCN method proposed by Chen et al. [37]. Our method is simple, fast, and has better performance.” Where are these test results ? If they are being compared it should be clear what is being compared.

Response:

Thank you to the reviewer for their suggestions. We have added test results to the paper, mainly comparing the classification performance of the test set under the same conditions with the training time of the training set. As shown in Table 7.

9. In section 3 it must be clarified how the data set is divided into train/test or train/validation/test and with what percentages. It is mentioned that:

- 70% of the total number of samples are used as the training set,

- 30% are used as the test set with 1503 data points (is this 30% of 5011? for 50x50 blocks, please clarify since there are 4 different blocks in table 1)

Furthermore, there is no reference to the validation set (is validation a part of the training set ? What percentage ? ) and the total amount of In fact the hyper-parameter tuning is mentioned in line 358 but not in the experimental results section. Without a clear procedure of train, hyper-parameter tuning, and test, with clearly defined datasets we may face biased performance results.

Table 2 and 3 refer to 50x50 blocks?

Table 3 and 4 How are these tests sets?

Figure 7 are not histograms. These are bar graphs and they do not add any value to the work. Well organized tables present the same results in an efficient way.

The discussion and conclusion sections should be “4.” and “4)” ?

The suggested size of data blocks is “50×50 for subsequent experiments”, but the performance (81.87% accuracy) is presented for 80x80 blocks. These two aspects should be coherent and present the performance of the suggested block.

Response:

Thank you to the reviewer for their suggestions.

(1) For each dataset of different sizes, the segmented training set accounts for 70% and the test set accounts for 30%. Among them, 1/7 of the training set is the validation set used to assist in training. When segmenting the training and testing sets, we strive to ensure a relative balance between the number of benign and malignant cases in the training set, while also ensuring that the data in the training and testing sets come from different cases, thereby reducing the correlation between the training and testing sets and facilitating the simulation of clinical needs. We believe that our segmentation method can effectively ensure the generalization performance of the model. But in order to improve the persuasiveness of our model, we conducted a ten-fold cross validation experiment on a dataset of size 50 x 50 x 196, randomly selecting the test set and training set in ten rounds (ensuring that the training set and test set come from different cases as much as possible), and finally calculated the average accuracy as shown in Table 5.

(2) We have labeled the size of the data blocks in the text, and the data sizes used in Tables 2 and 3 are 50 x 50 x 196.

(3) We have modified the bar chart to a line chart, which can more clearly show the changes in processing performance of datasets of different sizes, as shown in Figure 7.

(4) We have revised the description of the final conclusion based on the suggestions, making the logic of the article clearer and better presenting the performance of the suggested size.

(5) We have made extensive revisions to address language errors throughout the entire text to make the explanation clearer.

Round 2

Reviewer 2 Report

Comments and Suggestions for Authors

The authors have addressed all my concerns. The paper can be accepted for publication.

Author Response

Thank you for your review and evaluation. Your suggestions have provided us with important and constructive insights on this manuscript, and have greatly improved it. We have made sufficient revisions to the manuscript, striving to achieve a level that can be published.

Reviewer 3 Report

Comments and Suggestions for Authors

The authors have addressed in good faith all the pointed issues. Nevertheless, the available revised version  including the highlighted changed sections are confusing considering the removed/deleted parts are not highlighted.

Minor suggestion:

- line 512 where is "(with 1/7 randomly selected as the validation set),", it will be more clear if the validation set is referred also as a percentage to be coherent with the rest. e.g."(with 1/7 randomly selected as the validation set, which amounts to 10% of the total data),"  or some equivalent sentence.

Comments on the Quality of English Language

The available revised version  including the highlighted changed sections are confusing considering the removed/deleted parts are not highlighted.

Thus, it is not possible to verify if the revised version is correct in terms of language. e.g.  in line 449 "three metrics have been*are* used to" the *are* is clearly the highlighted new part, the "have been" is certainly to be deleted,  but is not marked to be deleted. This situation repeats in the entires revised version.

Author Response

Dear Reviewer:

Thank you very much for your kind and careful comments. We have fully considered all your comments and made substantial revisions to our manuscript based on your feedback. We have made every effort to improve the manuscript and hope that the revised version will be approved. The response is as follows:

1.The authors have addressed in good faith all the pointed issues. Nevertheless, the available revised version including the highlighted changed sections are confusing considering the removed/deleted parts are not highlighted.

Minor suggestion:

- line 512 where is "(with 1/7 randomly selected as the validation set),", it will be more clear if the validation set is referred also as a percentage to be coherent with the rest. e.g."(with 1/7 randomly selected as the validation set, which amounts to 10% of the total data),"  or some equivalent sentence.

Response:

We greatly appreciate the valuable feedback from the reviewers. In response to the language expression issues raised by the reviewers, we have made detailed revisions to the entire text, striving to make the language of the manuscript describe the research process more clearly. At the same time, we found that there was an issue where the deleted/ removed parts were not highlighted when generating the PDF version. So after making language modifications to the entire text, we regenerated the PDF version to ensure that all deleted/added parts were highlighted.

2.The available revised version including the highlighted changed sections are confusing considering the removed/deleted parts are not highlighted.

Thus, it is not possible to verify if the revised version is correct in terms of language. e.g.  in line 449 "three metrics have been*are* used to" the *are* is clearly the highlighted new part, the "have been" is certainly to be deleted,  but is not marked to be deleted. This situation repeats in the entires revised version.

Response:

Response:

Thank you for the reviewer's reminder. We have re-edited the document to ensure that all modified parts are highlighted.

We are indebted to you for your outstanding and constructive comments, which greatly helped us to improve the technical quality and presentation of our manuscript. Once again, thank you very much for your comments and suggestions.